# YAP1 activation by human papillomavirus E7 promotes basal cell identity in squamous epithelia

Joshua Hatterschide[1], Paola Castagnino[1], Hee Won Kim[1], Steven M Sperry[1†], Kathleen T Montone[2], Devraj Basu[1], Elizabeth A White[1]*

[1]Department of Otorhinolaryngology: Head and Neck Surgery, University of Pennsylvania Perelman School of Medicine, Philadelphia, United States; [2]Department of Pathology and Laboratory Medicine, University of Pennsylvania Perelman School of Medicine, Philadelphia, United States

*For correspondence: eawhite@pennmedicine.upenn.edu

Present address: †Department of Otolaryngology-Head and Neck Surgery, Aurora St. Luke's Medical Center, Milwaukee, United States

Competing interest: The authors declare that no competing interests exist.

**Abstract** Persistent human papillomavirus (HPV) infection of stratified squamous epithelial cells causes nearly 5% of cancer cases worldwide. HPV-positive oropharyngeal cancers harbor few mutations in the Hippo signaling pathway compared to HPV-negative cancers at the same anatomical site, prompting the hypothesis that an HPV-encoded protein inactivates the Hippo pathway and activates the Hippo effector yes-associated protein (YAP1). The HPV E7 oncoprotein is required for HPV infection and for HPV-mediated oncogenic transformation. We investigated the effects of HPV oncoproteins on YAP1 and found that E7 activates YAP1, promoting YAP1 nuclear localization in basal epithelial cells. YAP1 activation by HPV E7 required that E7 binds and degrades the tumor suppressor protein tyrosine phosphatase non-receptor type 14 (PTPN14). E7 required YAP1 transcriptional activity to extend the lifespan of primary keratinocytes, indicating that YAP1 activation contributes to E7 carcinogenic activity. Maintaining infection in basal cells is critical for HPV persistence, and here we demonstrate that YAP1 activation causes HPV E7 expressing cells to be retained in the basal compartment of stratified epithelia. We propose that YAP1 activation resulting from PTPN14 inactivation is an essential, targetable activity of the HPV E7 oncoprotein relevant to HPV infection and carcinogenesis.

## Editor's evaluation

The oncogenic virus Human Papillomavirus encodes the E7 protein which is an important contributor to carcinogenesis. The authors of this publication discovered a novel function of HPV E7, that contributes to its carcinogenic properties. They show that the ability of E7 to extend the lifespan of keratinocytes and facilitate basal cell retention are both activities mediated by the basal-cell specific activation of the cellular protein YAP1.

## Introduction

Human papillomaviruses (HPV) are nonenveloped viruses with circular double-stranded DNA genomes that infect keratinocytes in stratified squamous epithelia (*Doorbar et al., 2015*; *Graham, 2017*; *McBride, 2017*). Although most HPV infections are cleared by the immune system, some infections persist and form higher grade lesions that can lead to cancer (*Koshiol et al., 2008*; *McBride, 2022*; *Radley et al., 2016*; *Rositch et al., 2013*). HPV infection at mucosal epithelial sites causes cancers including oropharyngeal, cervical, vaginal, penile, and anal malignancies (*de Martel et al., 2017*;

**eLife digest** The 'epithelial' cells that cover our bodies are in a constant state of turnover. Every few weeks, the outermost layers die and are replaced by new cells from the layers below. For scientists, this raises a difficult question. Cells infected by human papillomaviruses, often known as HPV, can become cancerous over years or even decades. How do infected cells survive while the healthy cells around them mature and get replaced?

One clue could lie in PTPN14, a human protein which many papillomaviruses eliminate using their viral E7 protein; this mechanism could be essential for the virus to replicate and cause cancer. To find out the impact of losing PTPN14, Hatterschide et al. used human epithelial cells to make three-dimensional models of infected tissues. These experiments showed that, when papillomaviruses destroy PTPN14, a human protein called YAP1 turns on in the lowest, most long-lived layer of the tissue. Cells in which YAP1 is activated survive while those that carry the inactive version mature and die. This suggests that papillomaviruses turn on YAP1 to remain in tissues for long periods.

Papillomaviruses cause about five percent of all human cancers. Finding ways to stop them from activating YAP1 has the potential to prevent disease. Overall, the research by Hatterschide et al. also sheds light on other epithelial cancers which are not caused by viruses.

*Gillison et al., 2015*). Nearly 5% of human cancer cases are caused by persistent infection with one of the high-risk (oncogenic) HPV genotypes (*de Martel et al., 2020*).

Inactivation of host cell tumor suppressors by the high-risk HPV E6 and E7 oncoproteins modulates cellular processes that enable HPV persistence. Two well-characterized instances of tumor suppressor inactivation by HPV are high-risk HPV E6 proteins targeting p53 for proteasome-mediated degradation and high-risk HPV E7 proteins binding and degrading the retinoblastoma protein (RB1) (*Heck et al., 1992*; *Münger et al., 1989*; *Scheffner et al., 1990*; *Werness et al., 1990*). Both p53 degradation and RB1 inactivation are required for productive HPV infection (*Collins et al., 2005*; *Flores et al., 2000*; *Kho et al., 2013*; *McLaughlin-Drubin et al., 2005*; *Wang et al., 2009*). In addition to supporting productive infection, E7 is essential for HPV-mediated carcinogenesis (*Mirabello et al., 2017*). The impact of the HPV oncoproteins on cell growth control pathways is reflected in human cancer genomic data: genes in the p53 pathway and in the RB1-related cell cycle pathway are frequently mutated in HPV-negative head and neck squamous cell carcinoma (HNSCC) but infrequently mutated in HPV-positive HNSCC (*Sanchez-Vega et al., 2018*).

Although some of the growth-promoting activities of high-risk HPV E6 and E7 are well established, open questions remain. RB1 binding/degradation by high-risk HPV E7 is necessary but insufficient for E7 transforming activity (*Balsitis et al., 2006*; *Balsitis et al., 2005*; *Banks et al., 1990*; *Ciccolini et al., 1994*; *Helt and Galloway, 2001*; *Huh et al., 2005*; *Ibaraki et al., 1993*; *Jewers et al., 1992*; *Phelps et al., 1992*; *Strati and Lambert, 2007*; *White et al., 2015*). Papillomavirus researchers have sought to identify one or more activities of HPV E7 that cooperate with RB1 inactivation to promote carcinogenesis and to identify the cellular pathway affected by such an activity. Human cancer genomic data indicates that like the p53 and cell cycle pathways, the Hippo signaling pathway is more frequently mutated in HPV-negative than in HPV-positive HNSCC. The core Hippo pathway consists of a kinase cascade upstream of the effector proteins yes-associated protein (YAP1) and its paralog transcriptional coactivator with PDZ binding motif (TAZ). When the Hippo kinases are inactive, YAP1 and TAZ are activated and translocate to the nucleus. In stratified squamous epithelia, YAP1 is primarily expressed in the basal layer, where YAP1 activation is regulated by contextual cues including cell density, tension in the extracellular matrix, and contact with the basement membrane (*Elbediwy et al., 2016*; *Totaro et al., 2017*; *Zhang et al., 2011*). In normal stratified squamous epithelia, activation of YAP1 and TAZ promotes expansion of the basal cell compartment and inhibition of YAP1 and TAZ allows keratinocytes to differentiate (*Beverdam et al., 2013*; *Elbediwy and Thompson, 2018*; *Schlegelmilch et al., 2011*; *Totaro et al., 2017*; *Yuan et al., 2020*; *Zhang et al., 2011*). Mutations in many of the tumor suppressors upstream of YAP1/TAZ are common in a variety of cancer types (*Moroishi et al., 2015*).

Protein tyrosine phosphatase non-receptor type 14 (PTPN14) has been implicated as a tumor suppressor and negative regulator of YAP1 (*Knight et al., 2018*; *Mello et al., 2017*; *Poernbacher et al., 2012*; *Wang et al., 2012*). E7 proteins from diverse HPV genotypes bind directly to PTPN14

and recruit the E3 ligase UBR4 to direct PTPN14 for proteasome-mediated degradation (*Szalmás et al., 2017*; *White et al., 2016*; *White et al., 2012b*; *Yun et al., 2019*). We have shown that PTPN14 degradation and RB1 binding/degradation are separable activities of HPV E7 that each contributes to E7 carcinogenic activity (*Hatterschide et al., 2020*; *Hatterschide et al., 2019*; *White et al., 2016*). However, the downstream consequences of PTPN14 degradation are poorly understood, and so far we have not observed that PTPN14 inactivation in human keratinocytes causes an increase in canonical YAP1 target genes *CTGF* and *CYR61*.

These observations regarding an additional transforming activity of HPV E7, the ability of E7 to inactivate PTPN14, and the relative paucity of mutations in the Hippo pathway in HPV-positive HNSCC led us to hypothesize that HPV E7-mediated activation of YAP1 is required for the transforming activity of high-risk HPV E7. Here we show that expression of high-risk HPV E7 is sufficient to activate YAP1 and that HPV E7 requires YAP1/TAZ-TEAD transcriptional activity to promote cell growth. We demonstrate that HPV E7 must bind PTPN14 to activate YAP1 and that PTPN14 inactivation alone is sufficient to activate YAP1. YAP1 activation by HPV E7 is restricted to the basal layer of the epithelium where we found *PTPN14* expression to be enriched.

Our finding that either HPV E7 or PTPN14 loss activated YAP1 specifically in basal epithelial cells led us to investigate the role of YAP1 activation during normal HPV infection. HPV infection begins in basal epithelial keratinocytes (*Day and Schelhaas, 2014*; *Pyeon et al., 2009*; *Roberts et al., 2007*) and infected basal cells are the site of persistent HPV infection (*Doorbar et al., 2021*). The basal cell compartment contains the only long-lived cells in the epithelium and the HPV genome can be maintained in dividing cells in a largely dormant state (*Egawa et al., 2012*; *Parish et al., 2006*; *You et al., 2004*). Activation of YAP1 and TAZ has been proposed to maintain the progenitor cell state in several different epithelia (*Beverdam et al., 2013*; *Heng et al., 2020*; *Hicks-Berthet et al., 2021*; *Szymaniak et al., 2015*; *Yimlamai et al., 2014*; *Zhao et al., 2014*). If YAP1 activation by E7 promotes the maintenance of a basal cell state in stratified squamous epithelia, YAP1 activation could facilitate the persistence of HPV-positive cells. Testing this hypothesis, we found that YAP1 activation and PTPN14 degradation by E7 both promote the maintenance of cells in the basal compartment of stratified epithelia. We propose that YAP1 activation facilitates HPV persistence and contributes to the carcinogenic activity of high-risk HPV E7.

## Results

### HPV E7 activates YAP1 in basal keratinocytes

A comprehensive analysis of somatic mutations and copy number variations in human tumor samples revealed that the cell cycle, p53, and Hippo pathways are the three pathways that exhibit the greatest difference in alteration frequency in HPV-negative vs HPV-positive HNSCC (*Sanchez-Vega et al., 2018*). We used data made available by The Cancer Genome Atlas (TCGA) through cBioPortal (*Lawrence et al., 2015*) to recapitulate the finding that genes in these pathways are altered at a lower frequency in HPV-positive than in HPV-negative HNSCC (*Figure 1A* and *Figure 1—figure supplement 1*). However, most HPV-positive HNSCC arise in the oropharynx. We repeated the analysis of pathway alteration rates using data only from HPV-positive and HPV-negative oropharyngeal squamous cell carcinomas (OPSCC) (*Figure 1A* and *Figure 1—figure supplement 1*). Consistent with previous findings, HPV-negative OPSCC were more frequently altered in the p53, cell cycle, and Hippo pathways than HPV-positive OPSCC. Hippo pathway alterations in HPV-negative HNSCC or OPSCC include amplification of the YAP1/TAZ oncogenes or inactivating mutations in one or more of the upstream inhibitors of YAP1/TAZ, including PTPN14. Either alteration type is consistent with a carcinogenic role for YAP1 activation in HNSCC.

To test whether an HPV-encoded protein activates YAP1, we grew three dimensional (3D) organotypic epithelial cultures to model the differentiation of keratinocytes into basal and suprabasal compartments. Organotypic cultures of primary human foreskin keratinocytes (HFK) harboring an HPV18 genome exhibited increased YAP1 staining and increased YAP1 nuclear localization, indicative of YAP1 activation, in the basal layer of the epithelium compared to HFK cultures (*Figure 1B* and *Figure 1—figure supplement 2A-E*). Proliferating cell nuclear antigen (PCNA) transcription increases upon RB1 inactivation and is a marker of HPV E7 expression (*Cheng et al., 1995*; *Flores et al., 2000*; *Lee et al., 1995*). Although YAP1 activation in the HPV18 genome containing cells was restricted to

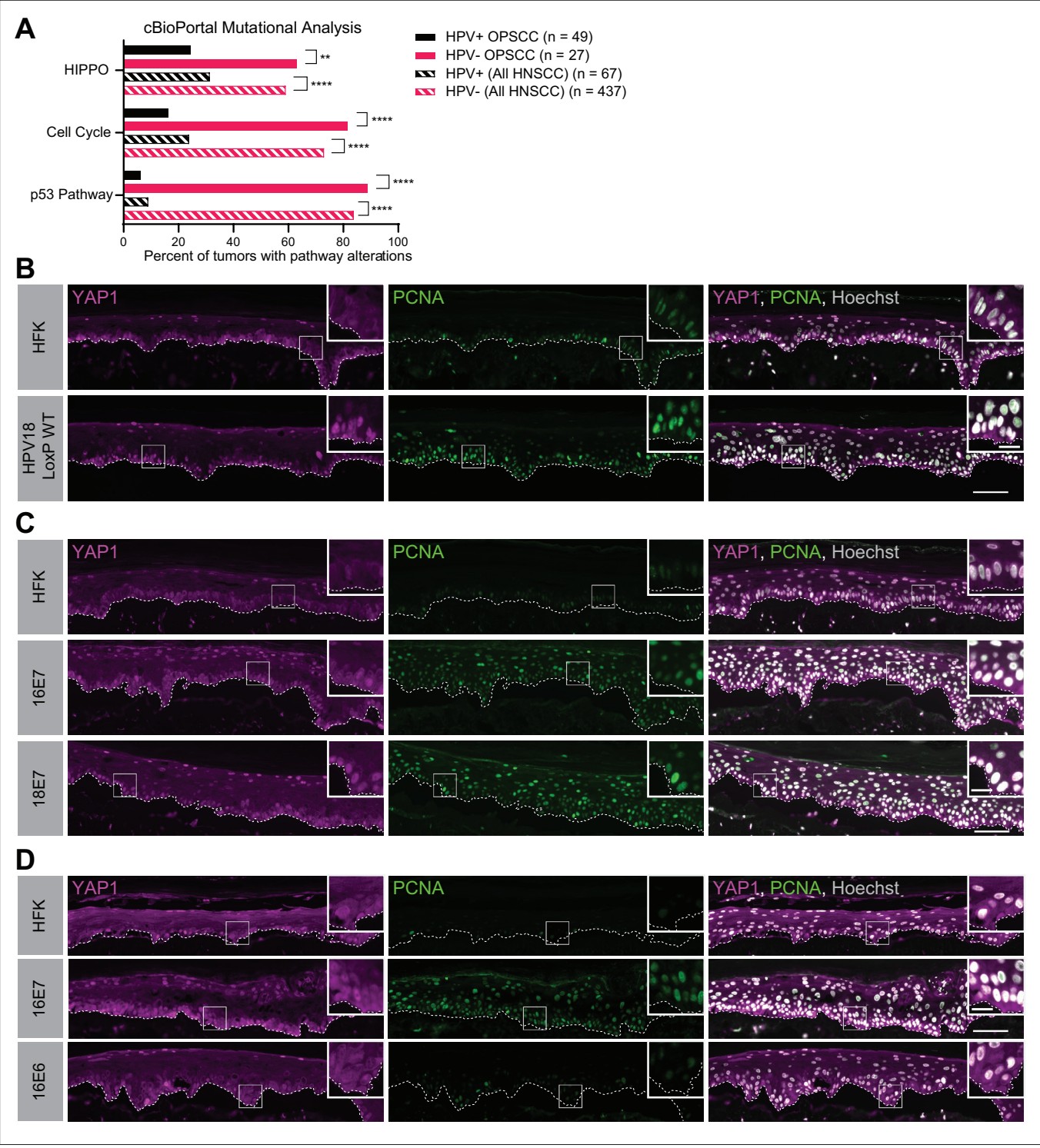

**Figure 1.** Human papillomavirus (HPV) E7 activates yes-associated protein (YAP1) in basal epithelial keratinocytes. (**A**) cBioPortal analysis for total genomic mutations and copy number alterations in HPV-positive and HPV-negative oropharyngeal squamous cell carcinoma (OPSCC) and head and neck squamous cell carcinoma (HNSCC). Graph displays the percent of tumors with alterations in each pathway. Statistical significance was determined by Fisher's exact test (**p<0.01, ****p<0.0001). (**B–D**) Organotypic cultures were grown from primary human foreskin keratinocytes (HFK), HFK harboring the HPV18 genome, or HFK transduced with retroviral expression vectors encoding HPV E6 or E7 proteins. Formalin-fixed paraffin-embedded sections of cultures grown from (**B**) HFK or HFK harboring the HPV18 genome, (**C**) HFK or HFK expressing HPV16 E7 or HPV18 E7, or (**D**) HFK or HFK expressing HPV16 E6 or HPV16 E7 were stained for YAP1 (magenta), proliferating cell nuclear antigen (PCNA) (green), and Hoechst (gray). White dashed lines

*Figure 1 continued on next page*

*Figure 1 continued*

indicate the basement membrane. White boxes indicate the location of insets in main images. Main image scale bars = 100 µm. Inset scale bars = 25 µm.

The online version of this article includes the following figure supplement(s) for figure 1:

**Figure supplement 1.** Human papillomavirus (HPV)-positive head and neck squamous cell carcinoma (HNSCC) have fewer Hippo pathway alterations than HPV-negative HNSCC.

**Figure supplement 2.** Human papillomavirus type 18 (HPV18) genomes activate yes-associated protein (YAP1) in basal keratinocytes.

**Figure supplement 3.** Human papillomavirus (HPV) E7 activates yes-associated protein (YAP1) in basal keratinocytes.

**Figure supplement 4.** Human papillomavirus (HPV) E6 does not activate yes-associated protein (YAP1) in basal keratinocytes.

**Figure supplement 5.** Quantification of yes-associated protein (YAP1) activation by human papillomavirus (HPV) E6 or E7 in basal keratinocytes.

the basal cells, PCNA levels were elevated in both the basal and suprabasal layers of the epithelium in the same cultures.

We next tested whether either high-risk HPV E6 or E7 alone was sufficient to activate YAP1. HFK transduced with retroviral expression vectors encoding HPV16 E6, HPV16 E7, or HPV18 E7 were used to grow organotypic cultures. YAP1 expression and nuclear localization were increased in the HPV16 E7 and HPV18 E7 expressing cells relative to parental HFK (*Figure 1C*, *Figure 1—figure supplement 3A-C*, and *Figure 1—figure supplement 5A-C*). Similar to our observation in the HPV18 genome-containing cells, YAP1 activation was restricted to the basal epithelial layer. YAP1 nuclear localization increased modestly in organotypic cultures of HPV16 E6 expressing cells (*Figure 1D*, *Figure 1—figure supplement 4*, and *Figure 1—figure supplement 5A-C*). Constitutive expression of either HPV16 E7 or HPV18 E7 induced PCNA expression in basal and suprabasal cells. We conclude that HPV promotes increased YAP1 expression and nuclear localization in basal keratinocytes and that E7 is sufficient for YAP1 activation.

## HPV E7 activates YAP1 in keratinocytes through PTPN14 degradation

We previously discovered that HPV E7 binds and targets the YAP1 inhibitor PTPN14 for proteasome-mediated degradation (*White et al., 2016*; *White et al., 2012b*). We tested whether loss of PTPN14 expression in keratinocytes was sufficient to activate YAP1 in stratified epithelia by growing 3D organotypic cultures from existing control and PTPN14 knockout (KO) N/Tert-Cas9 keratinocytes (sgPTPN14) (*Hatterschide et al., 2019*). We found that YAP1 levels and YAP1 nuclear localization were increased in PTPN14 KO cultures compared to controls (*Figure 2A* and *Figure 2—figure supplement 1A-F*). YAP1 activation in basal epithelial cells lacking PTPN14 was comparable to YAP1 activation in HPV E7 cells. We conclude that loss of PTPN14 expression activates YAP1 in basal keratinocytes.

A highly conserved C-terminal arginine in E7 makes a direct interaction with the C-terminus of PTPN14 and the HPV18 E7 R84S variant is unable to bind or degrade PTPN14 (*Hatterschide et al., 2020*; *Yun et al., 2019*). To test whether PTPN14 degradation by HPV E7 is required for activation of YAP1, we grew 3D organotypic cultures using primary HFK transduced with retroviral expression vectors encoding HPV18 E7 wild type (WT) or HPV18 E7 R84S. Indeed, YAP1 expression and nuclear localization in the basal layer of HPV18 E7 R84S cultures were reduced compared to HPV18 E7 WT controls (*Figure 2B*, *Figure 1—figure supplement 5A-C*, and *Figure 2—figure supplement 2*).

In addition to activating YAP1, PTPN14 loss increased basal cell density from an average of 5.5 cells per 100 µm in control cultures to 9.0 cells per 100 µm in PTPN14 KO cultures (*Figure 2C*). Basal cell density was higher in HPV18 E7 WT cultures (9.4 cells per 100 µm) than in HPV18 E7 R84S cultures (7.1 cells per 100 µm) (*Figure 2D*). No statistically significant difference in suprabasal cell density was observed in either comparison (*Figure 2E and F*). We conclude that E7 expression or PTPN14 loss in stratified squamous epithelia is sufficient to activate YAP1 in the basal layer of the epithelium and increase basal cell density.

## *PTPN14* expression is enriched in basal keratinocytes

YAP1 activation was restricted to basal epithelial cells in our organotypic cultures, leading us to hypothesize that PTPN14 may act as a basal layer specific inhibitor of YAP1. We therefore sought to determine whether *PTPN14* expression is restricted to a specific subset of cells in the stratified epithelium. In a recent single-cell RNA sequencing analysis of human neonatal foreskin epidermis, *PTPN14* mRNA

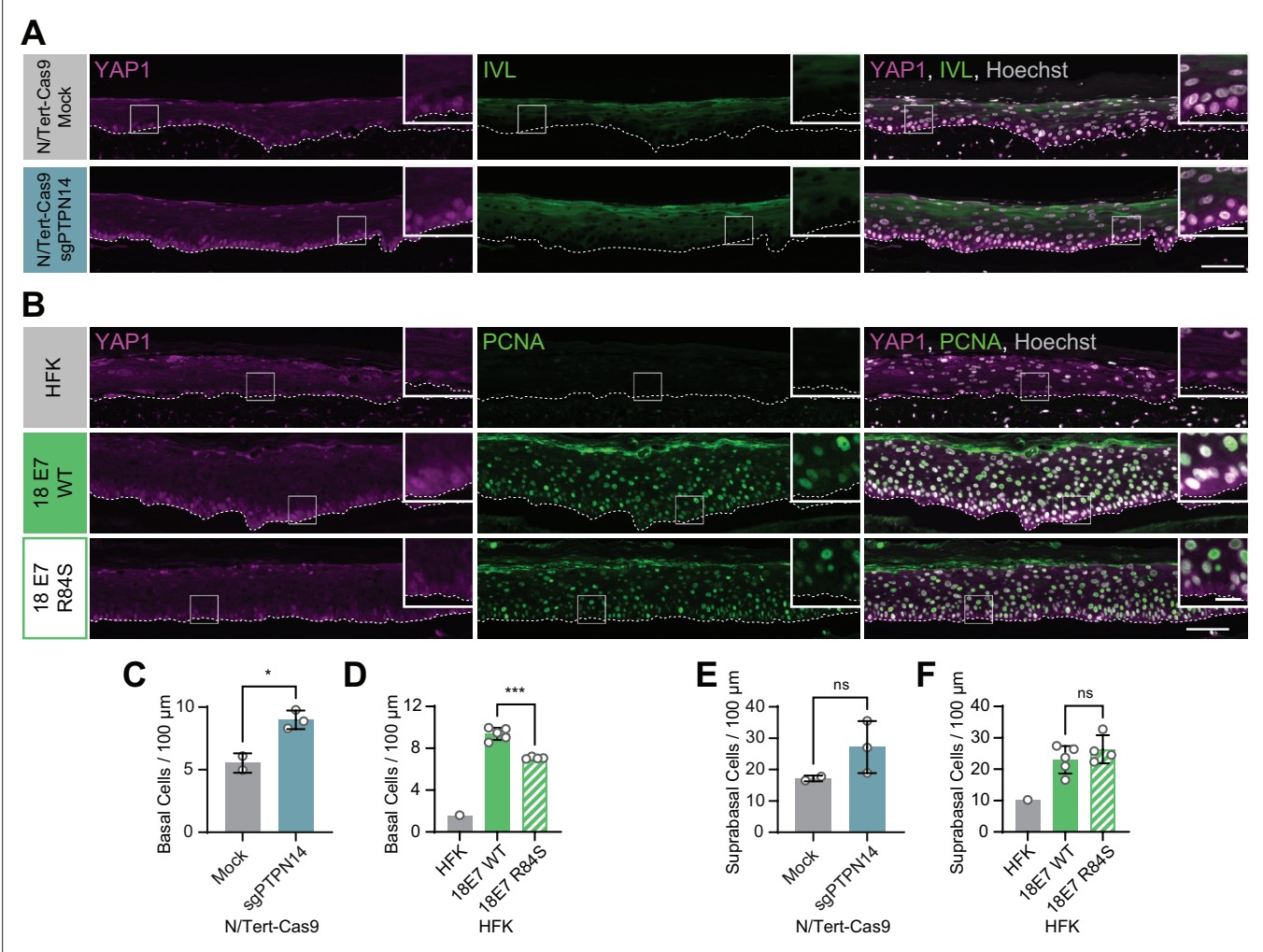

**Figure 2.** Human papillomavirus (HPV) E7 activates yes-associated protein (YAP1) in basal keratinocytes through protein tyrosine phosphatase non-receptor type 14 (PTPN14) degradation. Organotypic cultures were grown from N/Tert-Cas9 keratinocytes transfected with sgRNA or from primary human foreskin keratinocytes (HFK) transduced with retroviral expression vectors encoding HPV18 E7 WT or R84S. (**A**) Formalin-fixed paraffin-embedded (FFPE) sections of cultures grown from mock or sgPTPN14 transfected N/Tert-Cas9 keratinocytes were stained for YAP1 (magenta), involucrin (IVL) (green), and Hoechst (gray). (**B**) FFPE sections of cultures grown from parental HFK, HPV18 E7 WT, or HPV18 E7 R84S expressing HFK were stained for YAP1 (magenta), proliferating cell nuclear antigen (PCNA) (green), and Hoechst (gray). White dashed lines indicate the basement membrane. White boxes indicate the location of insets in main images. Main image scale bars = 100 μm. Inset scale bars = 25 μm. (**C–F**) Quantification of the number of (**C and D**) basal cells and (**E and F**) suprabasal cells per 100 μm of epidermis. Graphs display the mean ± SD and each individual data point (independent organotypic cultures). (**C and E**) Statistical significance was determined by t-test. (**D and F**) Statistical significance was determined by ANOVA using the Holm-Šídák correction for multiple comparisons (*p<0.05, ***p<0.001).

The online version of this article includes the following figure supplement(s) for figure 2:

**Figure supplement 1.** Protein tyrosine phosphatase non-receptor type 14 (PTPN14) knockout activates yes-associated protein (YAP1) in basal keratinocytes.

**Figure supplement 2.** Human papillomavirus (HPV) E7 activates yes-associated protein (YAP1) in basal keratinocytes through protein tyrosine phosphatase non-receptor type 14 degradation.

expression was enriched in the basal-III cluster (*Figure 3A and B*; *Wang et al., 2020a*). The basal-III cell cluster was described to be nonproliferating, marked by COL17A1 expression, and predicted to differentiate directly into spinous cells based on pseudotime analysis. *PTPN14* expression was higher in basal-III cells than in the spinous or granular cell clusters. To measure *PTPN14* expression in basal and suprabasal cells in our cultures, we used laser capture microdissection to isolate basal and suprabasal layers from 3D organotypic cultures grown from unmodified primary HFK (*Figure 3C*). We found that there was an approximately fivefold enrichment of *PTPN14* mRNA in the basal epithelial layer

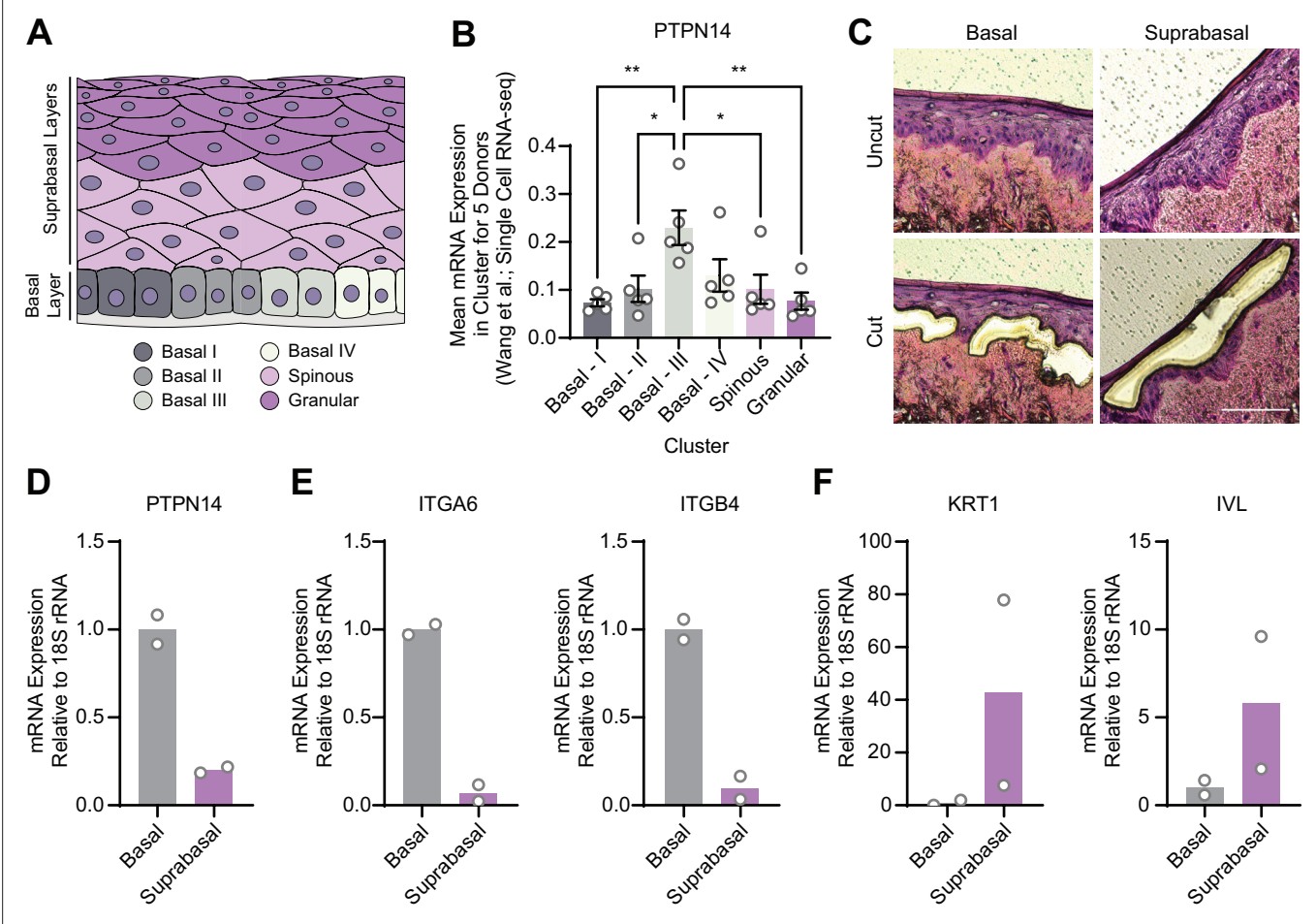

**Figure 3.** Protein tyrosine phosphatase non-receptor type 14 (PTPN14) expression is enriched in basal keratinocytes. (**A–B**) Single-cell RNA sequencing data and clustering analysis from Wang et al. was reanalyzed to assess PTPN14 expression in different subsets of epidermal cells. (**A**) Diagram of epidermis; shading depicts tissue localization of cell clusters. (**B**) For each donor, the mean of PTPN14 mRNA expression was calculated for each cell cluster. Graphs display the mean of PTPN14 mRNA expression for each donor (circles) as well as the mean of all five donors ± SEM (bars and error bars). Statistical significance was determined by ANOVA using the Holm-Sídák correction for multiple comparisons (*p<0.05, **p<0.01). (**C–F**) Basal and suprabasal layers from organotypic cultures were dissected using laser capture microdissection. (**C**) Representative images of HFK cultures before and after individual laser dissections. Hundreds of such cuts were performed per sample. (**D–F**) RNA was purified from isolated layers and qRT-PCR was used to assess the expression of PTPN14 (**D**), basal cell markers ITGA6 and ITGB4 (**E**), and differentiation markers KRT1 and IVL (**F**). Graphs display the mean and each individual data point.

The online version of this article includes the following figure supplement(s) for figure 3:

**Figure supplement 1.** Protein tyrosine phosphatase non-receptor type 14 (PTPN14) expression is enriched in basal keratinocytes in human papillomavirus type 18 (HPV18) E7 expressing organotypic cultures.

---

compared to the suprabasal layers (*Figure 3D*). As expected, the basal integrins *ITGA6* and *ITGB4* were expressed in the basal layer (*Figure 3E*) and the differentiation markers *KRT1* and *IVL* were expressed in the suprabasal layers (*Figure 3F*). The same pattern of *PTPN14* mRNA expression was observed in an organotypic culture grown from primary HFK expressing HPV18 E7 WT (*Figure 3— figure supplement 1A-C*). We conclude that *PTPN14* mRNA is enriched in basal keratinocytes in the presence or absence of HPV E7. Our data support that PTPN14 acts as a YAP1 inhibitor specifically in the basal compartment of stratified epithelia.

## YAP1/TAZ regulate differentiation downstream of PTPN14

In previous unbiased experiments we found that the primary effect of PTPN14 inactivation on transcription is to repress epithelial differentiation gene expression (*Hatterschide et al., 2020*; *Hatterschide et al., 2019*). However, we also observed that PTPN14 inactivation did not increase expression

of the canonical YAP1/TAZ targets *CTGF* and *CYR61*. Consistent with this difference there was minimal overlap between PTPN14-dependent differentially expressed genes and the genes listed in the MSigDB conserved YAP1 signature (*Figure 4A*). We therefore asked whether the ability of PTPN14 to regulate differentiation gene expression requires YAP1/TAZ as intermediates. Transduction of keratinocytes with a PTPN14 lentivirus induced the expression of the differentiation markers *KRT10* and *IVL* in a dose-dependent manner (*Figure 4—figure supplement 1A-C*). To test whether PTPN14 required YAP1/TAZ to increase differentiation marker gene expression, we transfected HFK with siRNAs targeting *YAP1* and *WWTR1* (the gene encoding TAZ) then transduced the cells with PTPN14 lentivirus (*Figure 4B*). HFK transfected with control siRNA exhibited the expected increase in *KRT1* and *IVL* after transduction with PTPN14 lentivirus (*Figure 4C and D* and *Figure 4—figure supplement 2A,B*). However, keratinocytes depleted of YAP1/TAZ did not express relatively more *KRT1* or *IVL* when *PTPN14* was overexpressed than when it was not. We conclude that PTPN14 requires YAP1 and/or TAZ to regulate differentiation gene expression in keratinocytes. Both pairs of YAP1/TAZ siRNA had the same effect on differentiation in response to *PTPN14* overexpression yet only one pair efficiently depleted TAZ protein levels (*Figure 4B*), leading us to speculate that YAP1 is the key intermediate connecting PTPN14 levels to differentiation gene expression.

Next, we tested whether repression of keratinocyte differentiation occurs upon loss of LATS1 and LATS2, the core Hippo pathway kinases that phosphorylate and inhibit YAP1 and TAZ. We used siRNAs to deplete *PTPN14*, *LATS1*, or *LATS2* and measured the expression of the differentiation markers *KRT1* and *IVL* (*Figure 4E and F*). Depletion of *PTPN14, LATS1,* or *LATS2* all decreased differentiation gene expression to a similar degree. None of the three knockdowns significantly affected the levels of *CTGF* or *CYR61* (*Figure 4G–H*). Direct depletion of *YAP1* or *WWTR1* affected both differentiation gene expression and *CTGF/CYR61* levels. Other than siWWTR1-08, all siRNAs used in these experiments reduced the expression of their target genes by twofold or greater (*Figure 4I*). *YAP1* knockdown always had a stronger effect than did *WWTR1* knockdown and our qRT-PCR analyses supported that *WWTR1* transcript levels were low in HFK. This result shows that inactivation of three different YAP1 inhibitors dampens differentiation gene expression even in the absence of a differentiation stimulus and does not increase canonical YAP1 target gene expression in keratinocytes. We have previously observed that *PTPN14* knockout or E7 expression reduces differentiation gene expression both in undifferentiated cells and in cells stimulated to differentiate (*Hatterschide et al., 2020*; *Hatterschide et al., 2019*). Taken together, these data support that PTPN14 promotes differentiation through inhibition of YAP1/TAZ despite not affecting canonical YAP1/TAZ target genes.

## HPV-positive HNSCC are less differentiated than HPV-negative HNSCC

We next asked whether the gene expression pattern observed downstream of PTPN14 loss is reflected in HPV-positive cancers. HPV-positive HNSCC have a strong propensity toward poorly differentiated, basaloid histology (*Mendelsohn et al., 2010*; *Pai and Westra, 2009*), which is reflected in their transcriptional profile (*Hatterschide et al., 2019*). We confirmed the relationship between HPV positivity and greater impairment of differentiation by immunohistochemical analysis of the differentiation marker KRT1 in sections of 14 HPV-negative tumors and 48 HPV-positive tumors (*Figure 5A*). Forty-three percent of HPV-negative tumors and 12.5% of HPV-positive tumors stained positive for KRT1. We also measured gene expression in patient-derived xenograft (PDX) models generated from human HNSCC. We measured *KRT1, KRT10,* and *IVL* levels using RNA extracted from 11 HPV-negative and 8 HPV-positive HNSCC PDX. Each differentiation marker was expressed at a lower level in HPV-positive PDX than in HPV-negative PDX (*Figure 5B*). We observed the same pattern of differentiation marker gene expression in an analysis of transcriptomic data from other cohorts (*Figure 5—figure supplement 1A-C*; *Lawrence et al., 2015*). Having confirmed that HPV-positive HNSCC exhibit reduced expression of differentiation markers compared to HPV-negative HNSCC, we measured *CTGF* and *CYR61* levels. We found no significant difference in expression of these canonical YAP1/TAZ target genes in HPV-positive vs HPV-negative PDX, although there was a trend toward higher *CTGF* in the HPV-positive PDX (*Figure 5C*). In the transcriptomic analyses, *CTGF* and *CYR61* expression trended higher in the HPV-negative HNSCC (*Figure 5—figure supplement 1D,E*). The pattern of low expression of differentiation markers and unchanged canonical YAP1/TAZ target gene expression in HPV-positive versus HPV-negative patient samples is consistent with the effects of PTPN14 inactivation in cultured cells.

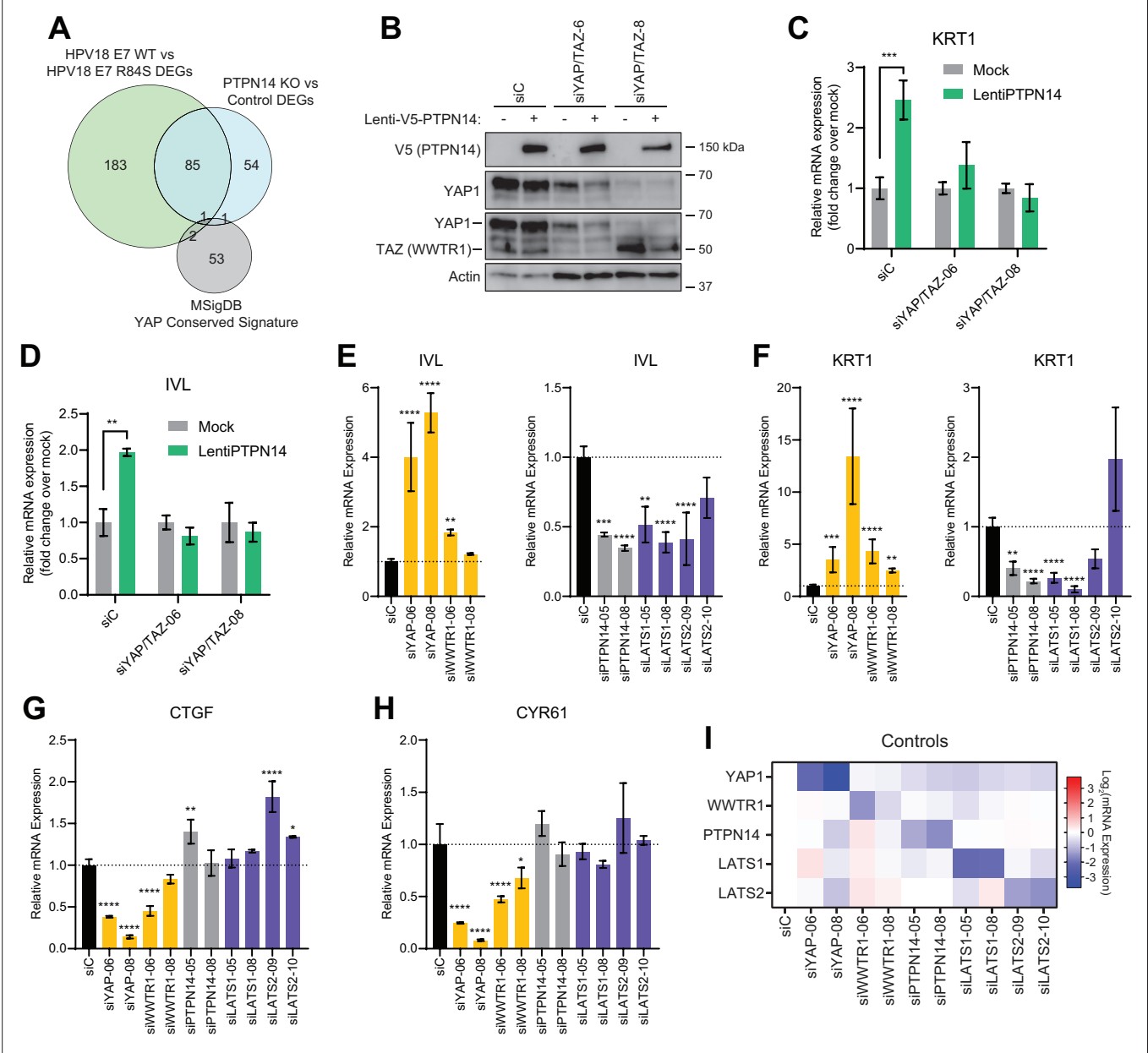

**Figure 4.** Yes-associated protein (YAP1)/TAZ regulate differentiation downstream of protein tyrosine phosphatase non-receptor type 14 (PTPN14). (**A**) Venn diagram comparing the MSigDB YAP conserved signature to the differentially expressed genes (DEG) from our two published experiments that reflect PTPN14 loss in keratinocytes. (**B–D**) *YAP1* and *WWTR1* were simultaneously knocked down by siRNA transfection in human foreskin keratinocytes (HFK). Transfected HFK were then transduced with PTPN14 lentivirus at 24 hr post-transfection. Cells were lysed for protein and total cellular RNA at 72 hr post-transfection. (**B**) Cell lysates were subjected to SDS-PAGE/Western blot analysis and probed with antibodies to PTPN14, YAP1, TAZ, and actin (*Figure 4—source data 1*). TAZ blot was generated by reprobing the membrane that was originally probed for YAP1. (**C and D**) qRT-PCR was used to measure the expression of the differentiation markers *KRT1* and *IVL* relative to *G6PD*. Graphs display fold change in gene expression relative to the mock transduced cells. (**E–I**) Primary HFK were transfected with siRNAs targeting *YAP1*, *WWTR1* (TAZ), *PTPN14*, *LATS1*, and *LATS2*. Two siRNAs were used per target. qRT-PCR was used to measure gene expression for: the differentiation markers *IVL* (**E**) and *KRT1* (**F**), and the canonical YAP1/TAZ targets *CTGF* (**G**) and *CYR61* (**H**). Data confirming that individual siRNA transfections depleted intended transcripts are summarized in a heatmap of $\log_2$(fold-change) levels (**I**). Bar graphs display the mean ± SD of three independent replicates. Statistical significance of each treatment compared to siC was determined by ANOVA using the Holm-Sídák correction for multiple comparisons (*p<0.05, **p<0.01, ***p<0.001, ****p<0.0001).

The online version of this article includes the following source data and figure supplement(s) for figure 4:

**Source data 1.** Original images for Western blots in *Figure 4B*.

**Figure supplement 1.** Protein tyrosine phosphatase non-receptor type 14 (PTPN14) overexpression promotes differentiation in keratinocytes.

*Figure 4 continued on next page*

Figure 4 continued

**Figure supplement 1—source data 1.** Original images for Western blots in *Figure 4—figure supplement 1A*.

**Figure supplement 2.** Yes-associated protein (YAP1) and TAZ are required for protein tyrosine phosphatase non-receptor type 14 (PTPN14) to promote keratinocyte differentiation.

## High-risk HPV E7 require YAP1/TAZ-TEAD transcriptional activity to extend the lifespan of primary keratinocytes

High-risk but not low-risk HPV E7 proteins can extend the lifespan of primary keratinocytes (*Halbert et al., 1991*). The TEADi protein is a genetically encoded competitive inhibitor that prevents binding between YAP1/TAZ and TEAD transcription factors (*Yuan et al., 2020*). We used TEADi to test whether YAP1/TAZ-TEAD transcriptional activity was required for high-risk HPV E7 to extend the lifespan of primary HFK. We transduced HFK with retroviral vectors encoding GFP, HPV16 E7, or HPV18 E7 plus a lentiviral vector encoding doxycycline-inducible GFP-TEADi. As anticipated, HPV16 E7 or HPV18 E7 extended the lifespan of primary HFK based on cumulative population doublings (*Figure 6A and B*). TEADi induction upon doxycycline treatment decreased the lifespan of primary HFK in the presence or absence of E7, but the effect of YAP1/TAZ-TEAD inhibition was greater in the HPV16 E7 and HPV18 E7 cells, where E7 had minimal ability to promote growth in the presence of TEADi. We conclude that high-risk HPV E7 proteins require YAP1/TAZ-TEAD transcriptional activity for their lifespan extending capacity in primary keratinocytes.

## PTPN14 loss and YAP1 activation promote basal cell retention in organotypic cultures

YAP1 overexpression impairs differentiation and promotes progenitor cell identity in squamous and nonsquamous epithelia. HPV infection is maintained in a reservoir of infected basal cells and productive virus replication begins upon commitment to differentiation. To better understand how repression of differentiation downstream of YAP1 activation affects HPV viral biology, we developed an assay to measure cell retention in the basal epithelial layer. We hypothesized that YAP1 activation by HPV E7 might promote the adoption or maintenance of a basal cell identity in stratified squamous epithelia. In our cell fate monitoring assay, a small proportion of GFP-labeled cells were mixed with unmodified parental HFK and the pool was used to generate organotypic cultures in which normal labeled cells are randomly distributed throughout the epithelium.

Our initial experiment tested whether YAP1 activation altered cell fate in stratified squamous epithelia. We used GFP-labeled tracing cells that expressed doxycycline-inducible YAP1 WT, YAP1 S127A (hyperactive), or YAP1 S94 (cannot bind TEAD transcription factors) (*Figure 7—figure supplement 1A,B*). In organotypic cultures grown from a 1:25 mixture of GFP-labeled cells and unmodified HFK, about 20% of uninduced GFP+ cells were found in the basal layer. Induction of YAP1 WT or YAP1 S127A expression was sufficient to promote the retention of nearly 60% of labeled cells in the basal layer of the epithelium (*Figure 7A and B*). Only around 40% of GFP+ cells were found in the basal layer when YAP1 S94A was induced. These data indicate that YAP1 activation causes cells to be retained in the basal layer of a stratified squamous epithelium. The ability of YAP1 to bind TEAD transcription factors contributed to its activity in the cell fate assay.

We next tested whether loss of PTPN14 expression was sufficient to promote basal cell identity. We grew organotypic cultures from mixtures of unmodified primary HFK and GFP-labeled control or PTPN14 KO HFK (*Figure 7—figure supplement 1C, D*). 60%–70% of PTPN14 KO tracer cells were found in the basal layer when either of two PTPN14 guide RNAs were used whereas about 20% of control tracer cells were retained in the basal layer (*Figure 7C and D*). Thus, PTPN14 knockout is sufficient to promote basal cell fate determination in keratinocytes.

Next, we tested whether HPV E7 promoted basal cell retention and if so, whether its cell retention activity required PTPN14 degradation. We grew organotypic cultures from mixtures of GFP-labeled HFK expressing HPV18 E7 WT, HPV18 E7 R84S, or the empty vector control diluted 1:50 into unmodified primary HFK (*Figure 7—figure supplement 1E, F*). We found that nearly 80% of GFP-labeled HPV18 E7 WT tracer cells were retained in the basal layer compared to about 10% of labeled control cells (*Figure 7E and F*). HPV18 E7 WT labeled cells were numerous and grouped in clusters in the basal layer, suggesting that E7 promoted the clonal expansion of labeled basal cells. Both effects

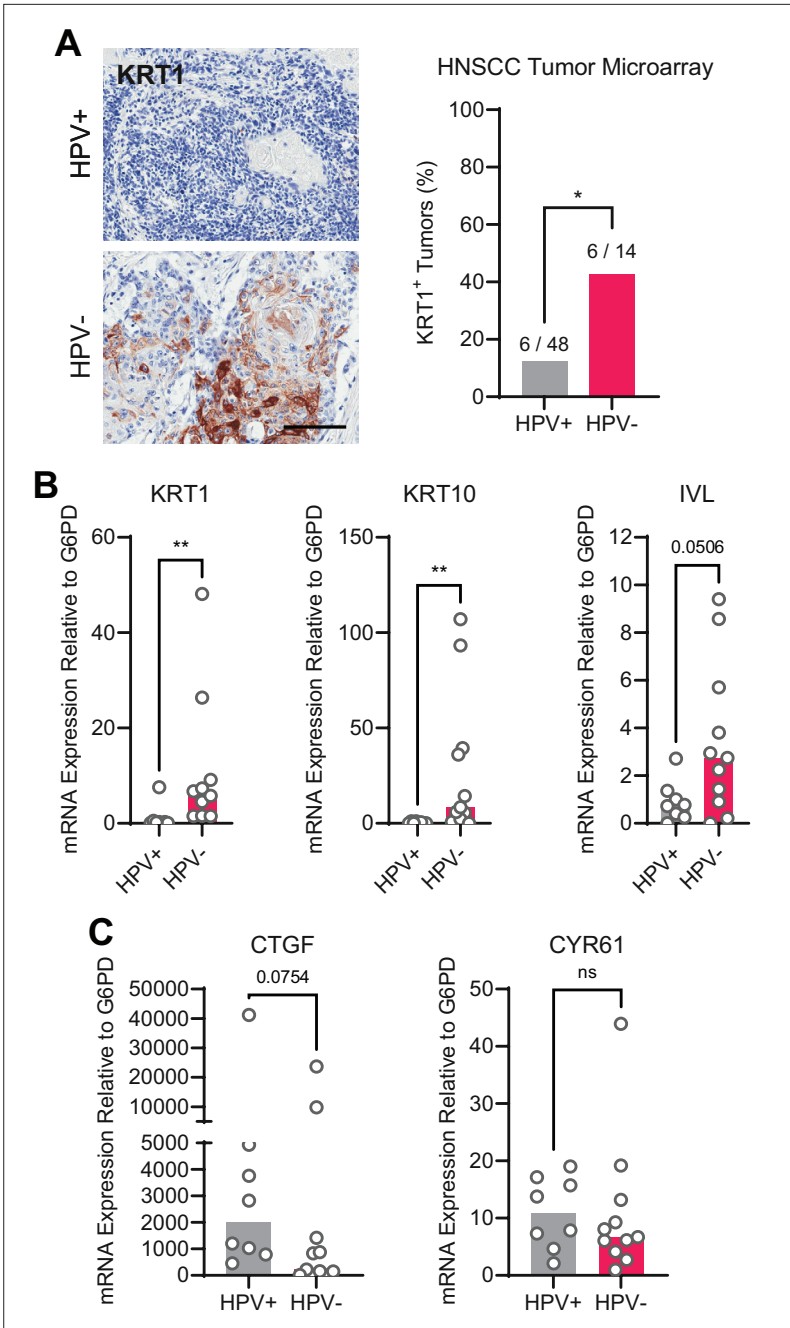

**Figure 5.** Human papillomavirus (HPV)-positive head and neck squamous cell carcinoma (HNSCC) are less differentiated than HPV-negative HNSCC. (**A**) Human HNSCC tumor samples were stained for KRT1 (left). Scale bar = 100 μm. Graph displays the percentage of tumors that were KRT1+ (right). Statistical significance was determined by Fisher's exact test. (**B–C**) Total RNA was purified from patient-derived xenograft samples and qRT-PCR was used to assess gene expression of (**B**) the differentiation markers KRT1, KRT10, and IVL and (**C**) the canonical yes-associated protein 1/TAZ targets CTGF and CYR61. Statistical significance was determined by Mann-Whitney nonparametric test (*p<0.05, **p<0.01).

The online version of this article includes the following figure supplement(s) for figure 5:

**Figure supplement 1.** Human papillomavirus (HPV)-positive head and neck squamous cell carcinomas (HNSCC) expresse lower levels of differentiation genes than HPV-negative HSNCC.

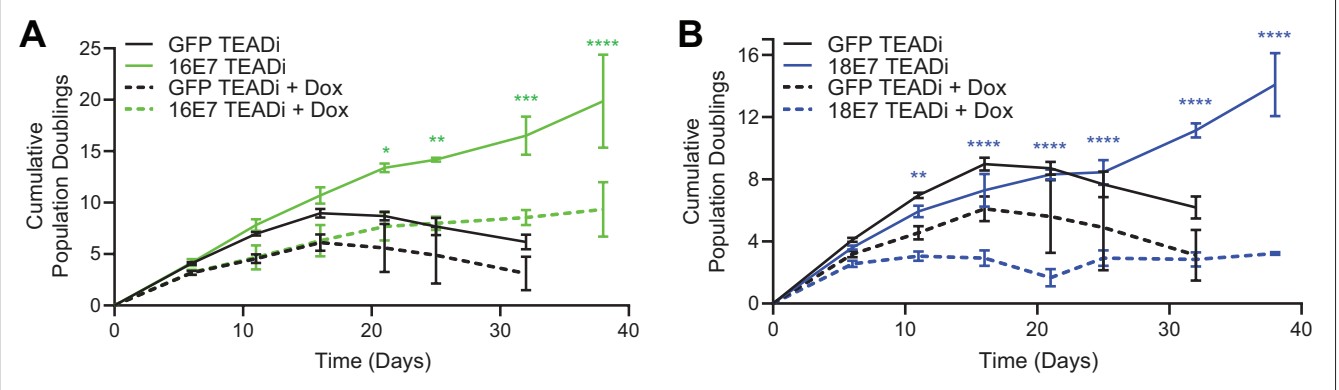

**Figure 6.** High-risk human papillomavirus (HPV) E7 requires yes-associated protein 1/TAZ-TEAD transcriptional activity to extend the lifespan of primary keratinocytes. Primary human foreskin keratinocytes (HFK) were transduced with retroviruses encoding HPV16 E7, HPV18 E7, or GFP, plus pInducer20 TEADi lentivirus. Each cell population was cultured with or without 1 µg/mL doxycycline (dox) in the media for 38 days and population doublings were tracked with each passage. Graph displays the mean ± SD of two independently transduced cell populations per condition. Statistical significance when comparing cell growth with and without doxycycline was determined by two-way ANOVA using the Sídák correction for multiple comparisons (*p<0.05, **p<0.01, ***p<0.001, ****p<0.0001).

were dampened in experiments using HPV18 E7 R84S tracer cells (cannot degrade PTPN14). Labeled HPV18 E7 R84S cells exhibited varying degrees of basal cell expansion and basal cell retention and approximately 60% of labeled cells were in the basal layer. HPV18 E7 R84S retains the ability to inactivate RB1 and we interpret these data to mean that the proliferation of labeled basal cells resulted from RB1 inactivation. Finally, HPV18 E7 ΔDLLC cannot bind RB1 but can bind and degrade PTPN14. In a cell fate experiment using GFP-labeled HPV18 E7 ΔDLLC tracer cells, the labeled cells were present mainly as single cells in the basal layer (*Figure 7—figure supplement 2A-B*). The behavior of the two mutant HPV E7 proteins supports that PTPN14 degradation is required for basal cell retention and RB1 inactivation is required for basal cell expansion. We conclude that PTPN14 degradation and YAP1 activation by HPV18 E7 promote basal cell retention.

## Discussion

YAP1 and TAZ are oncogenes that promote growth and inhibit differentiation in stratified squamous epithelia (*Elbediwy et al., 2016*; *Schlegelmilch et al., 2011*; *Totaro et al., 2017*; *Yuan et al., 2020*; *Zhang et al., 2011*). Here we report that HPV E7 activates YAP1 (*Figure 1*). YAP1/TAZ-TEAD transcriptional activity is required for the carcinogenic activity of HPV E7 (*Figure 6*) and YAP1 activation by E7 biases HPV E7-expressing cells to be retained in the basal epithelial layer (*Figure 7*). Based on these findings we propose that YAP1 activation by HPV E7 enables HPV-infected cells to persist in stratified epithelia. There is substantial evidence that RB1 inactivation is necessary but insufficient for the transforming activity of high-risk HPV E7 (*Balsitis et al., 2006*; *Balsitis et al., 2005*; *Banks et al., 1990*; *Ciccolini et al., 1994*; *Helt and Galloway, 2001*; *Huh et al., 2005*; *Ibaraki et al., 1993*; *Jewers et al., 1992*; *Phelps et al., 1992*; *Strati and Lambert, 2007*; *White et al., 2015*). We propose that YAP1 activation cooperates with RB1 inactivation to enable the transforming activity of HPV E7.

PTPN14 binding by HPV18 E7 was required for activation of YAP1 in the basal layer and PTPN14 KO was sufficient for the same effect (*Figure 2*). Highly conserved amino acids in E7 participate in binding to PTPN14 (*Hatterschide et al., 2020*; *Yun et al., 2019*), indicating that YAP1 activation and maintenance of basal cell state is likely shared among diverse papillomavirus E7 proteins. Some minor genotype-specific differences were apparent. HPV18 E7 depletes PTPN14 protein levels more efficiently than HPV16 E7 (*Hatterschide et al., 2020*; *White et al., 2016*), which is consistent with the observed stronger effect of HPV18 E7 on YAP1 nuclear localization in basal cells (*Figure 1*). Genotype-specific differences could also explain the stronger effect of TEADi on HPV18 E7 in lifespan extension assays (*Figure 6*). Although other reports have suggested that HPV might activate YAP1 (*He et al., 2015*; *Morgan et al., 2020*; *Olmedo-Nieva et al., 2020*; *Webb Strickland et al., 2018*), no specific activity of an HPV protein has previously been shown to enable YAP1 activation. Other groups have

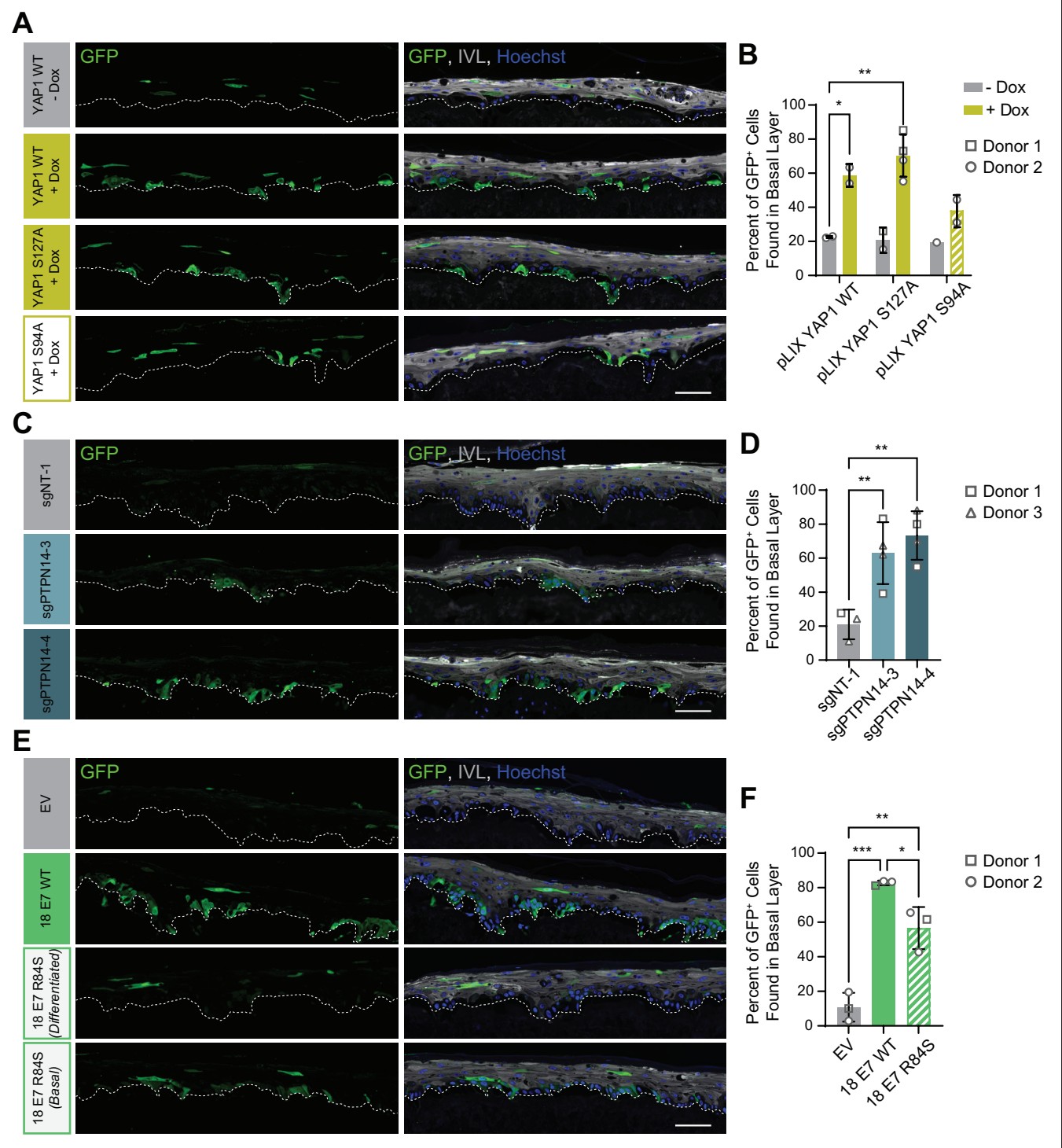

**Figure 7.** Protein tyrosine phosphatase non-receptor type 14 (PTPN14) loss and yes-associated protein (YAP1) activation by human papillomavirus (HPV) E7 promote basal cell retention in organotypic cultures. Organotypic cultures were grown from GFP-labeled human foreskin keratinocytes (HFK) mixed with unmodified HFK. (**A–B**) GFP-labeled HFK were transduced with lentiviral vectors encoding YAP1 WT, YAP1 S127A, or YAP1 S94A under the control of a doxycycline (dox) inducible promoter. GFP-labeled YAP1 cells were mixed 1:25 into unmodified HFK and organotypic cultures were grown from the mixture. Cultures were grown ±1 µg/mL doxycycline. (**C–D**) GFP-labeled HFK were transduced with LentiCRISPR v2 vectors encoding control or PTPN14 targeting sgRNAs. GFP-labeled cells were mixed 1:25 into unmodified HFK and organotypic cultures were grown from the mixture. (**E–F**) GFP-labeled HFK were transduced with HPV18 E7 WT, HPV18 E7 R84S, or the empty vector (EV). GFP-labeled HPV18 E7 cells were mixed 1:50 into unmodified HFK and organotypic cultures were grown from the mixture. Two different images for 18E7 R84S reflect cases in which all tracer cells migrated to the

*Figure 7 continued on next page*

*Figure 7 continued*

suprabasal layers (*differentiated*) or in which some tracer cells remained in the basal layer (*basal*). (**A, C, E**) Formalin-fixed paraffin-embedded sections of cultures were stained for GFP (green), IVL (gray), and Hoechst (blue). Scale bar = 100 μm. (**B, D, F**) Quantification of the percentage of GFP+ cells found in the basal layer. Graphs display the mean ± SD and each individual data point (independent organotypic cultures). Shapes indicate cultures grown from different HFK donors. Statistical significance was determined by ANOVA using the Holm-Sídák correction for multiple comparisons (*$p < 0.05$, **$p < 0.01$).

The online version of this article includes the following source data and figure supplement(s) for figure 7:

**Figure supplement 1.** Protein tyrosine phosphatase non-receptor type 14 (PTPN14) degradation by human papillomavirus (HPV) E7 promotes basal cell retention.

**Figure supplement 1—source data 1.** Original images for Western blots in *Figure 7—figure supplement 1D*.

**Figure supplement 1—source data 2.** Original images for Western blots in *Figure 7—figure supplement 1F*.

**Figure supplement 2.** Human papillomavirus type 18 (HPV18) E7 can promote basal cell retention in the absence of retinoblastoma 1 protein binding.

proposed that HPV E6 activates YAP1 (*He et al., 2015*; *Webb Strickland et al., 2018*), and we observed modest YAP1 activation by HPV E6 compared to E7. We conclude that activation of YAP1 by HPV E7 is contingent upon its ability to bind and degrade PTPN14.

Even when HPV E7 was expressed in all layers of a stratified epithelium, YAP1 levels and nuclear localization increased only in basal epithelial cells. We found that E7 required PTPN14 degradation to activate YAP1 and that PTPN14 was expressed predominantly in basal keratinocytes (*Figure 3*). Basal cell-specific expression of *PTPN14* is consistent with the observation that it is regulated by p63, the master regulator of basal cell identity in stratified epithelia (*Perez et al., 2007*). We propose that PTPN14 inhibits YAP1 primarily in basal cells and that unlike the effects of E7 on RB1 in both differentiated and undifferentiated cells, E7 activates YAP1 primarily in basal cells.

Degradation of PTPN14 by HPV E7 represses keratinocyte differentiation but does not induce canonical Hippo pathway target genes (*Hatterschide et al., 2020*; *Hatterschide et al., 2019*). Nonetheless, we found that PTPN14 overexpression required YAP1/TAZ to promote differentiation gene expression (*Figure 4C and D*). Few studies have tested how YAP1 inhibitor inactivation alters gene expression downstream of YAP1. Here we demonstrate that inactivation of LATS1 or LATS2, two well-characterized inhibitors of YAP1/TAZ, also decreased expression of differentiation genes in unstimulated cells but did not induce canonical YAP1/TAZ targets (*Figure 4E–I*). Taken together, these experiments indicate that PTPN14 acts through YAP1/TAZ to regulate differentiation in keratinocytes. It is so far unclear why *CTGF* and *CYR61* expression is sensitive to large changes in total levels of YAP1 or TAZ yet is unaffected by alterations in regulators upstream of YAP1/TAZ. Nonetheless, the pattern of low differentiation gene expression and unchanged expression of canonical YAP1/TAZ target genes caused by PTPN14 loss is consistent with gene expression differences between HPV-positive and HPV-negative HNSCC.

PTPN14 knockout and knockdown reduced differentiation gene expression in monolayer culture. Even so, we did not observe reduced differentiation in suprabasal layers of organotypic cultures grown from PTPN14 knockout cells (*Figure 2A* and *Figure 2—figure supplement 1A-C*). Using our cell fate monitoring assay, we determined that instead, HPV18 E7 promotes basal cell retention and that either YAP1 overexpression or PTPN14 KO is sufficient for this activity (*Figure 7*). The HPV18 E7 R84S mutant that cannot degrade PTPN14 was impaired, but not completely deficient, in its ability to promote basal cell retention. We interpret these data to mean that E7 activities in addition to PTPN14 degradation might contribute to basal cell retention. GFP-labeled basal cells in the HPV18 E7 R84S cell fate monitoring experiments were present in clusters, suggesting that cells that were able to remain in the basal layer had undergone clonal expansion. It is possible that RB1 inactivation by the mutant E7 drives an apparent increase in R84S mutant basal cells in this assay. The effect of YAP1 activation on cell fate in our assay resembles several experiments in which YAP1 promotes progenitor cell identity in airway and liver epithelia (*Yimlamai et al., 2014*; *Zhao et al., 2014*). Our findings demonstrate that YAP1 activation enables basal cell fate determination in stratified squamous epithelia and show that loss of an inhibitor of YAP1 has the same effect. We conclude that one consequence of YAP1 activation by HPV E7 is that E7-expressing cells are retained in the basal layer of stratified squamous epithelia.

Although persistent infection is a prerequisite for HPV-mediated carcinogenesis, the mechanisms used by papillomaviruses to establish persistent infections remain incompletely understood. Maintaining infection in the basal cell compartment is critical for papillomavirus persistence. Substantial effort has been devoted to the mechanistic understanding of how the papillomavirus genome is stably maintained in the basal layer upon cell division. However, much less is known about how papillomaviruses manipulate epithelial cell fate to establish and expand the pool of infected basal cells. Previously, HPV E7 was believed to be primarily required to establish a cellular environment conducive to HPV DNA replication in suprabasal cells (*Cheng et al., 1995*; *Collins et al., 2005*; *Flores et al., 2000*; *McLaughlin-Drubin et al., 2005*). We propose that a so far unappreciated role of E7 is that it activates YAP1 to facilitate HPV persistence by biasing infected cells to remain in the basal layer of the epithelium. Not every HPV E7-expressing cell was retained in the basal layer, so we do not anticipate that YAP1 activation would block differentiation-dependent HPV replication. HPV E6 also represses differentiation gene expression in keratinocytes and has been proposed to promote basal cell retention (*Kranjec et al., 2017*). Further research is needed to determine the extent to which different HPV genotypes depend on the activities of E6 or E7 for basal cell retention activity.

To the best of our knowledge, no other viruses are recognized to modulate cell fate decisions in solid tissues in a way that facilitates persistence. Some herpesviruses impact the choice between progenitor/differentiated cell fates in infected immune cells, for example Epstein-Barr virus (EBV) and human herpesvirus 6B (HHV6B) restrict differentiation in infected cells (*Knox and Carrigan, 1992*; *Niiya et al., 2006*; *Onnis et al., 2012*; *Romeo et al., 2019*; *Styles et al., 2017*). Herpesviruses, polyomaviruses, and hepadnaviruses encode proteins proposed to activate YAP1/TAZ or alter Hippo signaling (*Hwang et al., 2014*; *Liu et al., 2015a*; *Liu et al., 2015b*; *Nguyen et al., 2014*; *Shanzer et al., 2015*; *Tian et al., 2004*; *Wang et al., 2019*). Not all of the mechanisms used by these viruses to activate YAP1 nor the downstream consequences of YAP1 activation have been well defined. Our finding that HPV E7 activates YAP1 to manipulate cell fate opens up an exciting new line of inquiry into how YAP1, TAZ, and the Hippo signaling pathway could impact viral infections by regulating tissue developmental processes.

YAP1 activation and PTPN14 are relevant to both viral and nonviral cancers. We found that a genetically encoded inhibitor of YAP1/TAZ-TEAD transcription inhibited the growth of high-risk HPV E7 expressing cells (*Figure 6*), indicating that high-risk HPV E7 proteins require YAP1 or TAZ for carcinogenesis. YAP1/TAZ activation is sufficient to drive carcinogenesis in mouse models of cervical and oral cancer (*He et al., 2019*; *Nishio et al., 2020*; *Omori et al., 2020*), and the YAP1 inhibitor verteporfin reduced the growth of HPV-positive tumors in a xenograft model (*Liu et al., 2019*). YAP1 activation correlates with the clinical stage of HPV infection (*Nishio et al., 2020*), and YAP1 localizes to the nucleus in HPV-positive cancers (*Alzahrani et al., 2017*). Basal cell carcinoma (BCC) is the nonviral cancer that is most clearly linked to PTPN14. Germline inactivating mutations in *PTPN14* are associated with a fourfold to eightfold increase in risk of BCC by age 70 (*Olafsdottir et al., 2021*) and somatic mutations in *PTPN14* are frequent in BCC (*Bonilla et al., 2016*). YAP1/TAZ-TEAD transcriptional activity also restricts differentiation in BCC cells (*Yuan et al., 2021*). We propose that the specific association of PTPN14 with BCC is related to our observation that PTPN14 loss activates YAP1 in basal epithelial cells. YAP1 inhibition is of major clinical interest for several cancer types, and it is appealing to speculate that targeting YAP1 could treat persistent HPV infection and/or HPV-positive cancers.

## Materials and methods

### Key resources table

| Reagent type (species) or resource | Designation | Source or reference | Identifiers | Additional information |
|---|---|---|---|---|
| Antibody | anti-Actin (Mouse monoclonal) | Sigma-Aldrich | Cat#: MAB1501; RRID:AB_2223041 | WB (1:20,000) |
| Antibody | anti-GFP (Rabbit polyclonal) | Invitrogen | Cat#: A6455; RRID:AB_221570 | WB (1:1,000); IHC-P (1:2000) |
| Antibody | anti-Mouse IgG Alexa Fluor 488 (Goat polyclonal) | Invitrogen | Cat#: A11001; RRID:AB_2534069 | IHC-P (1:250) |
| Antibody | anti-Mouse IgG HRP (Horse monoclonal) | Cell Signaling Technologies | Cat#: 7076; RRID:AB_330924 | WB (1:2000) |

*Continued on next page*

*Continued*

| Reagent type (species) or resource | Designation | Source or reference | Identifiers | Additional information |
|---|---|---|---|---|
| Antibody | anti-Rabbit IgG Alexa Fluor 594 (Goat polyclonal) | Invitrogen | Cat#: A11012; RRID:AB_2534079 | IHC-P (1:250) |
| Antibody | anti-Rabbit IgG HRP (Goat monoclonal) | Cell Signaling Technologies | Cat#: 7074; RRID:AB_2099233 | WB (1:2000) |
| Antibody | anti-HA-Peroxidase (Rat monoclonal) | Roche | Cat#: 12013819001; RRID:AB_390917 | WB (1:500) |
| Antibody | anti-ITGB4 (Rabbit polyclonal) | Sigma-Aldrich | Cat#: HPA036348; RRID:AB_2675077 | IHC-P (1:100) |
| Antibody | anti-IVL (Mouse monoclonal) | Santa Cruz Biotechnology | Cat#: sc-398952 | IHC-P (1:100) |
| Antibody | anti-KRT1 (Mouse monoclonal) | Enzo Life Sciences | Cat#: C34904; RRID:AB_2265594 | |
| Antibody | anti-PCNA | Santa Cruz Biotechnology | Cat#: sc-56; RRID:AB_628110 | IHC-P (1:100) |
| Antibody | Anti-PTPN14 (Rabbit monoclonal) | Cell Signaling Technology | D5T6Y; Cat#: 13808; RRID:AB_2798318 | WB (1:500) |
| Antibody | anti-TAZ (Rabbit monoclonal) | Cell Signaling Technology | D3I6D; Cat#: 70148; RRID:AB_2799776 | WB (1:1000) |
| Antibody | anti-V5 (Mouse monoclonal) | Invitrogen | Cat#: 46–0705 | WB (1:1000) |
| Antibody | anti-YAP1 (Rabbit monoclonal) | Cell Signaling Technology | D8H1X; Cat#: 14074; RRID:AB_2650491 | WB (1:1000); IHC-P (1:50) |
| Transfected construct (human) | nontargeting siRNA | Dharmacon | Cat#: D-001810–01 | |
| Transfected construct (human) | siRNA to YAP1 (OnTarget Plus) | Dharmacon | Cat#: J-012200–06 | |
| Transfected construct (human) | siRNA to YAP1 (OnTarget Plus) | Dharmacon | Cat#: J-012200–08 | |
| Transfected construct (human) | siRNA to WWTR1 (OnTarget Plus) | Dharmacon | Cat#: J-016083–06 | |
| Transfected construct (human) | siRNA to WWTR1 (OnTarget Plus) | Dharmacon | Cat#: J-016083–08 | |
| Transfected construct (human) | siRNA to PTPN14 (OnTarget Plus) | Dharmacon | Cat#: J-008509–05 | |
| Transfected construct (human) | siRNA to PTPN14 (OnTarget Plus) | Dharmacon | Cat#: J-008509–08 | |
| Transfected construct (human) | siRNA to LATS1 (OnTarget Plus) | Dharmacon | Cat#: J-004632–05 | |
| Transfected construct (human) | siRNA to LATS1 (OnTarget Plus) | Dharmacon | Cat#: J-004632–08 | |
| Transfected construct (human) | siRNA to LATS2 (OnTarget Plus) | Dharmacon | Cat#: J-003865–09 | |
| Transfected construct (human) | siRNA to LATS2 (OnTarget Plus) | Dharmacon | Cat#: J-003865–10 | |

## Plasmids and cloning

pInducer20 EGFP-TEADi was a gift from Ramiro Iglesias-Bartolome (Addgene plasmid # 140145) (*Yuan et al., 2020*). pQCXIH-Myc-YAP (Addgene plasmid # 33091), pQCXIH-Flag-YAP-S127A (Addgene plasmid # 33092), and pQCXIH-Myc-YAP-S94A (Addgene plasmid # 33094) were gifts from Kun-Liang Guan (*Zhao et al., 2007*). Each YAP1 ORF was amplified by PCR from pQCXIH, cloned into pDONR223, and transferred into pLIX_402 lentiviral backbone using Gateway recombination. pLIX_402 was a gift from David Root (Addgene plasmid # 41394). pLenti CMV GFP Hygro (656-4) was a gift from Eric Campeau & Paul Kaufman (Addgene plasmid # 17446) (*Campeau et al., 2009*). pHAGE-P-CMVt N-HA GFP was previously described (*Galligan et al., 2015*). pNeo-loxP-HPV18 was the kind gift of Thomas Broker and Louise Chow (*Wang et al., 2009*). The ΔDLLC mutation was introduced into the pDONR HPV18 E7 vector using site-directed mutagenesis. HPV18 E7 ΔDLLC and GFP ORFs were cloned into MSCV-P C-FlagHA GAW or MSCV-Neo C-HA GAW destination vectors using Gateway recombination. The remaining MSCV-P C-FlagHA and MSCV-Neo C-HA HPV E6 and HPV E7 retroviral plasmids and pHAGE lentiviral plasmids have been previously described (*Hatterschide et al., 2020*; *White et al., 2016*; *White et al., 2012a*; *White et al., 2012b*). A complete list of all plasmids used in this study is in *Supplementary file 1*.

## Cell culture, retrovirus production, and lentivirus production

Deidentified primary HFK and human foreskin fibroblasts (HFF) were provided by the University of Pennsylvania Skin Biology and Diseases Resource-based Center (SBDRC). N/Tert-1 cells are hTert-immortalized HFK (*Dickson et al., 2000*), and N/Tert-Cas9 mock and sgPTPN14-1 are N/Tert-1 cells further engineered to constitutively express Cas9 (*Hatterschide et al., 2019*). Keratinocytes for cell fate experiments were cultured in keratinocyte serum-free media (KSFM) (Life Technologies, Carlsbad, California) mixed 1:1 with Medium 154 (Thermo Fisher Scientific, Waltham, Massachusetts) with the human keratinocyte growth supplement (HKGS) (Thermo Fisher Scientific) (*Ridky et al., 2010*). Keratinocytes for all other experiments were cultured as *White et al., 2012a*. SBDRC-sourced primary HFK (*Figures 1–4 and 6*, *Figure 7*, *Figure 1—figure supplements 2–5*, *Figure 2—figure supplements 1 and 2*, *Figure 3—figure supplement 1*, *Figure 4—figure supplement 2*, and *Figure 7—figure supplements 1 and 2*) were derived from independent donors and were used within the first 2–8 passages after isolation. N/Tert-1 keratinocytes (*Figure 2A* and *Figure 4—figure supplement 1*) were obtained directly from the laboratory in which they were derived (*Dickson et al., 2000*) and were used in the first 2–4 passages post thawing. HFK are a short-lived source material that are not routinely STR profiled or tested for mycoplasma contamination. N/Tert-1 keratinocytes were observed at every passage to ensure that they retained keratinocyte morphology, displayed the cobblestone growth pattern characteristic of human skin keratinocytes, and were immortalized (grew indefinitely past the time at which they were used in the experiments reported herein). These cells also express keratinocyte specific genes (e.g. KRT1 and IVL) and retain the capacity to stratify and differentiate in organotypic culture. HFF were cultured in Dulbecco's Modified Eagle Medium (DMEM) (Thermo Fisher Scientific) supplemented with antibiotic and antimycotic. HFK harboring the HPV18 genome were previously described (*Hatterschide et al., 2020*), and were generated by transfecting cells with the pNeo-loxP-HPV18 vector (*Wang et al., 2009*) along with NLS-Cre and selecting with G418 to generate a stable population. Lentiviruses and retroviruses were produced in 293T or 293 Phoenix cells respectively as previously described (*White et al., 2016*). Stable keratinocyte populations were generated following transduction by selection with puromycin, G418, or hygromycin alone or in combination.

## Lifespan extension assay

Primary HFK were engineered and cultured as described in cell culture, retrovirus production, and lentivirus production. The growth of engineered HFK was monitored in culture for 38 days. Population doublings were calculated using the number of cells at the beginning and end of each passage.

## Organotypic epithelial culture

Devitalized human dermis was provided as deidentified material from the University of Pennsylvania SBDRC. Stands for organotypic epithelial cultures were printed using high temperature, autoclavable resin at the University of Pennsylvania Biotech Commons 3D-printing facility. Organotypic cultures were generated as previously described (*Duperret et al., 2015*; *Egolf et al., 2019*). Devitalized dermis was seeded with primary HFF on the dermal side at a density of $3 \times 10^4$ cells per cm$^2$ of culturing area and cultured for four days. Dermis and fibroblasts were then stretched across 3D-printed stands. The epidermal side of the dermis was seeded with unmodified or engineered keratinocytes at a density of $1 \times 10^6$ cells per cm$^2$. Organotypic cultures were cultured in E media (*Fehrmann and Laimins, 2005*) with the dermal layer maintained at the air-liquid interface starting on the day of seeding keratinocytes. Cultures were allowed to stratify for 12–14 days, then trimmed and fixed in 10% neutral buffer formalin for 24 hr. Tissues were embedded in paraffin and sectioned by the SBDRC Core A. A complete list of all organotypic cultures used in this study is in *Supplementary file 2*.

## siRNA transfection

Primary HFK were transfected with siRNAs using the Dharmafect 1 transfection reagent. All siRNA experiments were collected 72 hr post-transfection. Two siRNAs were used to target each gene in an experiment. The siRNAs used in this study were all purchased from Dharmacon (Lafayette, Colorado): nontargeting siRNA, siYAP1-06, siYAP1-08, siWWTR1-06, siWWTR1-08, siPTPN14-05, siPTPN14-08, siLATS1-05, siLATS1-08, siLATS2-09, siLATS2-10.

## Laser capture microdissection

Formalin-fixed paraffin-embedded (FFPE) organotypic cultures were sectioned onto polyethylene naphthalate membrane glass slides by the SBDRC Core A. Laser capture microdissection was performed on a Leica LMD 7000 microscope. Hundreds of microdissections were made per sample amounting to ~1.5 mm$^2$ of total dissected area per sample. RNA was isolated using the RNeasy FFPE kit (Qiagen, Germantown, Maryland). RNA concentration was determined using Qubit RNA HS assay kit (Life Technologies).

## Patient-derived xenografts

The PDXs were previously established from surgical resections of treatment-naive HPV-positive OPSCC as described (*Facompre et al., 2020*). Human tumors were engrafted subcutaneously in NSG mice and passaged at least twice before cryopreservation when they reached a volume of 0.5–1.0 cm$^3$. Total tumor RNA was isolated using the QIAamp RNA Blood Mini Kit (Qiagen).

## Western blotting

Western blots were performed using Mini-PROTEAN (Bio-Rad Laboratories, Hercules, California) or Criterion (Bio-Rad) Tris/Glycine SDS-PAGE gels and transfers were performed onto polyvinylidene difluoride. Membranes were blocked with 5% nonfat dry milk in Tris-buffered saline with 0.05% Tween 20 (TBST). Membranes were incubated with primary antibodies as specified in *Supplementary file 1*. Following TBST washes, membranes were incubated with horseradish peroxidase-coupled secondary antibodies and imaged using chemiluminescent substrate on an Amersham Imager 600 (GE Health-care, Chicago, Illinois).

## qRT-PCR

Unless otherwise specified, total cellular RNA was isolated using the NucleoSpin RNA extraction kit (Macherey-Nagel/Takara, San Jose, California). cDNA was generated from bulk RNA with the high-capacity cDNA reverse transcription kit (Applied Biosystems, Waltham, Massachusetts). cDNAs were used as a template for qPCR using Fast SYBR green master mix (Applied Biosystems) and a Quant-Studio 3 system (Thermo Fisher Scientific). 18S rRNA qRT-PCR primers were ordered from Integrated DNA Technologies (Integrated DNA Technologies, Inc, Coralville, Iowa): FWD, 5- CGCCGCTAGAGG TGAAATTCT; REV, 5- CGAACCTCCGACTTTCGTTCT (*Roh et al., 2005*). KiCqStart SYBR green primers for qRT-PCR (MilliporeSigma, St. Louis, Missouri) were used for the remaining genes assayed in this study: KRT1, KRT10, IVL, ITGB4, ITGA6, CYR61, CTGF, PTPN14, YAP1, WWTR1, LATS1, LATS2, G6PD, and GAPDH.

## Immunofluorescence, immunohistochemistry, and microscopy

FFPE sections were prepared for immunofluorescence by deparaffinization with xylene washes, rehydration through an ethanol gradient, and heat induced epitope retrieval (HIER). Tissue sections were blocked with PBS containing 1% bovine serum albumin, 10% normal goat serum, and 0.3% Triton X-100. Tissue sections were incubated with primary antibodies at 4°C overnight, washed with PBS with 0.05% Tween 20, and incubated with fluorescently labeled secondary antibodies and Hoechst 33342 at room temperature. Antibody dilutions and HIER conditions are specified in *Supplementary file 1*. Fluorescent micrographs were captured using an Olympus IX81 microscope. All fluorescent micrograph images within the same figure panels were captured using the same exposure time and batch processed using the same contrast settings.

YAP1 localization in basal epithelial cells was quantified by visual classification of the YAP1 staining in each cell as predominantly nuclear, predominantly cytoplasmic, or comparably distributed between the nucleus and cytoplasm using Hoechst stain to demarcate the nucleus. Basal or suprabasal cell identity in the cell fate monitoring assays was determined by visual classification based on costaining of GFP+ with the suprabasal cell marker IVL. All image analysis was performed using deidentified images with codified names.

The tissue microarray (TMA) was constructed from surgical resection specimens of 120 HNSCC that vary by TNM stage and HPV status (*Supplementary file 3*). Archival FFPE tumors of the oral cavity and oropharynx were identified retrospectively and oropharyngeal tumors were evaluated for HPV status as per College of American Pathologists criteria (*Lewis et al., 2018*) using IHC for p16. When present,

lymph node metastases were included in association with the primary tumor of origin. All FFPE specimens were represented in the TMA by at least three tissue cores that incorporate both non-necrotic central tumor regions and invasive margins. Staining for KRT1 was performed by the Clinical Services Laboratory in the University of Pennsylvania Department of Pathology and Laboratory Medicine. Antibody information can be found in *Supplementary file 1*. The KRT1 stained slides were reviewed with a standard light microscope, and evaluation was based on the presence or absence of staining in the cytoplasm of tumor cells.

## Bioinformatic analysis

Genomic mutation and copy number variation data as well as tumor RNA-seq gene expression data from TCGA (*Lawrence et al., 2015*) were analyzed using the cBioPortal.org graphical interface (*Cerami et al., 2012*; *Gao et al., 2013*). RNA-seq V2 RSEM (RNA-Seq by Expectation Maximization) normalized expression values for individual genes were downloaded directly from cBioPortal.org. OPSCC were distinguished from HNSCC by clinical annotation of primary tumor site and HPV-positive and HPV-negative status was assigned based on previously reported HPV transcript status (*Chakravarthy et al., 2016*). Genes included in each pathway analysis are listed in *Supplementary file 4*. Missense, truncating, and splice mutations of unknown significance as well as amplifications of tumor suppressor genes and deletion of oncogenes were excluded from total alteration tallies.

Single cell-RNA sequencing dataset derived from the human neonatal foreskin epidermis and subsequent clustering analysis were retrieved from GitHub (*Wang et al., 2020a*; *Wang et al., 2020b*) and reanalyzed with MATLAB. PTPN14 expression was calculated by averaging mRNA expression for all cells by cluster and donor.

## Acknowledgements

We thank the members of our laboratories, particularly Pavithra Rajagopalan, for helpful discussions. We thank Stephen M Prouty, Ph.D. from the SBDRC for help with tissue processing and sectioning. Stands for organotypic cultures were printed courtesy of the University of Pennsylvania Libraries' Biotech Commons.

## Additional information

### Funding

| Funder | Grant reference number | Author |
| --- | --- | --- |
| National Institute of Allergy and Infectious Diseases | T32 AI007324 | Joshua Hatterschide |
| National Institute of Dental and Craniofacial Research | F31 DE030365 | Joshua Hatterschide |
| American Cancer Society | 131661-RSG-18-048-01-MPC | Joshua Hatterschide Paola Castagnino Hee Won Kim Elizabeth A White |
| National Institute of Allergy and Infectious Diseases | R01 AI148431 | Joshua Hatterschide Paola Castagnino Hee Won Kim Elizabeth A White |
| National Institute of Dental and Craniofacial Research | R01 DE027185 | Devraj Basu |

The funders had no role in study design, data collection and interpretation, or the decision to submit the work for publication.

### Author contributions

Joshua Hatterschide, Conceptualization, Data curation, Formal analysis, Investigation, Methodology, Validation, Visualization, Writing - original draft, Writing - review and editing; Paola Castagnino,

Data curation, Investigation; Hee Won Kim, Investigation; Steven M Sperry, Resources; Kathleen T Montone, Formal analysis, Resources; Devraj Basu, Funding acquisition, Methodology, Resources, Writing - review and editing; Elizabeth A White, Conceptualization, Data curation, Formal analysis, Funding acquisition, Investigation, Project administration, Resources, Supervision, Writing - original draft, Writing - review and editing

**Author ORCIDs**
Joshua Hatterschide ⓘ http://orcid.org/0000-0003-3562-4166
Hee Won Kim ⓘ http://orcid.org/0000-0001-5642-0969
Elizabeth A White ⓘ http://orcid.org/0000-0001-7378-7690

**Ethics**
All patient-derived materials and clinical data in this study were obtained from patients who underwent surgery to remove an oral cavity or oropharyngeal cancer. Patients were counseled preoperatively and provided informed consent under University of Pennsylvania IRB-approved protocol #417200 "Head and Neck Cancer Specimen Bank" (PI: D. Basu) by signing a combined informed consent and HIPAA form for use of tissue for research. Consent under this longstanding and currently active protocol explicitly provides permission to access surgically removed fresh tumor tissue that is not needed for pathologic analysis as well as to access FFPE tumor tissue in the pathology archive at a later date. It also provides explicit permission to publish deidentified analyses of these resources. Patient care is not altered under this protocol, which carries minimal risk. Minors and other vulnerable populations are not included in the study.

**Decision letter and Author response**
Decision letter https://doi.org/10.7554/eLife.75466.sa1
Author response https://doi.org/10.7554/eLife.75466.sa2

## Additional files

**Supplementary files**
• Supplementary file 1. Plasmids and antibodies used in this study. Sheet one includes a list of all plasmids used in this study including details on the encoded genes, sgRNAs, promoters, antibiotic resistance markers, epitope tags, tag locations, selectable markers as well as Addgene plasmid numbers and citations of the original sources. Sheet two includes a list of all antibodies used in this study including details on the company, product number, and experimental conditions in which they were used.

• Supplementary file 2. Organotypic cultures used in this study. Lists all organotypic cultures analyzed in this study and includes details on the expresion vectors employed, the cell backgrounds used, and all figures and panels that portray each culture.

• Supplementary file 3. Tumor microarray specimen information. Lists the characteristics of the tumors included in the tumor microarray including the primary tumor site, HPV-positive/negative status, primary tumor T-stage, presence of nodal metastasis, and overall pathological stage.

• Supplementary file 4. Gene lists used for pathway mutational analyses. Lists the genes included in each pathway used in the pathway mutational analyses.

• Transparent reporting form

**Data availability**
Original, uncropped Western blot images are contained in Figure 4B - Source Data 1, Figure 4 supplement 1 - Source Data 1, Figure 7 supplement 1D - Source Data 1, Figure 7 supplement 1F - Source Data 1.

The following previously published datasets were used:

| Author(s) | Year | Dataset title | Dataset URL | Database and Identifier |
|---|---|---|---|---|
| Cerami E, Gao J, Dogrusoz U, Gross BE, Sumer SO, Aksoy BA, Jacobsen A, Byrne CJ, Heuer ML, Larsson E, Antipin Y, Reva B, Goldberg AP, Sander C, Schultz N | 2012 | The cBio Cancer Genomics Portal | https://www.cbioportal.org | cbioportal, cbioportal |

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
