## [Editor Report]

The oncogenic virus Human Papillomavirus encodes the E7 protein which is an important contributor to carcinogenesis. The authors of this publication discovered a novel function of HPV E7, that contributes to its carcinogenic properties. They show that the ability of E7 to extend the lifespan of keratinocytes and facilitate basal cell retention are both activities mediated by the basal-cell specific activation of the cellular protein YAP1.

---

## [Decision Letter]

**Decision letter after peer review:**

Thank you for submitting your article "YAP1 activation by human papillomavirus E7 promotes basal cell identity in squamous epithelia" for consideration by *eLife*. Your article has been reviewed by 2 peer reviewers, and the evaluation has been overseen by a Reviewing Editor and Sara Sawyer as the Senior Editor. The following individual involved in review of your submission has agreed to reveal their identity: Megan E Spurgeon (Reviewer #2).

Essential revisions:

Both reviewers were very positive about your manuscript and raised only a few points that we would like you to consider.

1. Please quantify the IF images.

2. Please adjust the points raised by reviewer #2 regarding figure quality and interpretation.

*Reviewer #1 (Recommendations for the authors):*

This is an excellent manuscript and I have very little criticism other than the point described in Public Review.

*Reviewer #2 (Recommendations for the authors):*

Overall, I was impressed by the data and multiple approaches used by the authors in this study. Aside from the weaknesses addressed above with respect to interpretations, I believe the results presented are compelling and support the authors' conclusions. Some of the recommendations below are provided to improve and/or clarify presentation. Others are scientific/experimental queries that, if considered and/or performed, would serve to strengthen the findings presented here but may understandably be outside the scope of this study.

1. Figure 1A: Was PTPN14 identified in the TCGA analysis as an inhibitor of YAP1/TAZ pathway often mutated in HPV-negative OPSCCs? If so, this could be noted.

2. Figure 1: There is no Part E in the figure but is indicated in the Legend. In Figure 1D, the PCNA staining is very difficult to see, even in the insets.

3. Figure 2 parts C-F: It would be easier to compare the effect on basal vs. suprabasal cells if the basal graphs were next to each other and suprabasal graphs were similarly presented side-by-side.

4. The authors found that E7 activates YAP1 primarily in basal cells. Has this been explored in 2D culture under conditions that induce differentiation? Does YAP1 activation decrease upon differentiation in these 2D experiments like what is seen in rafts? Such data would further strengthen their findings that YAP1 activation is specific to basal cells.

5. Regarding 'persistence': Have the authors explored whether a mutant HPV18 genome lacking E7 can replicate and/or be maintained over several passages in PTPN14KO cells? Likewise, does a genome carrying the R84S mutation persist in keratinocytes? These experiments would help provide key evidence to support the role of E7-mediated PTPN14 degradation/YAP1 activation in HPV persistence/maintenance.

6. Figure 3A/B: Can the authors provide any additional information about what differentiates the Basal Cell III population from I, II, and IV? In other words, are there other gene expression patterns that define the PTPN14-enriched basal cells?

7. Figure 6: were statistical comparisons performed?

8. Line 200-201: is it KRT10 or KRT1 that is being analyzed? Line 200 indicates KRT10, but line 201 indicates KRT1.

9. Citations needed:

a. Line 131: statement that PCNA1 increase upon RB1 inactivation

b. Line 375-376: "previously, HPVE7 was believed to be primarily required to establish a cellular environment conducive to HPV DNA replication in suprabasal cells"

---

## [Author Response]

Essential revisions:Both reviewers were very positive about your manuscript and raised only a few points that we would like you to consider.1. Please quantify the IF images.

Please see new Figure 1—figure supplement 5A-C and a detailed description below for the quantification of these images, which is now included in the manuscript.

2. Please adjust the points raised by reviewer #2 regarding figure quality and interpretation.

Please see new text on lines 373-379 and elsewhere as noted in the detailed descriptions below.

Reviewer #2 (Recommendations for the authors):Overall, I was impressed by the data and multiple approaches used by the authors in this study. Aside from the weaknesses addressed above with respect to interpretations, I believe the results presented are compelling and support the authors' conclusions. Some of the recommendations below are provided to improve and/or clarify presentation. Others are scientific/experimental queries that, if considered and/or performed, would serve to strengthen the findings presented here but may understandably be outside the scope of this study.1. Figure 1A: Was PTPN14 identified in the TCGA analysis as an inhibitor of YAP1/TAZ pathway often mutated in HPV-negative OPSCCs? If so, this could be noted.

We hypothesize that *PTPN14* encodes one of several cellular YAP1/TAZ inhibitors and that mutation in many different genes (not only *PTPN14*) could activate YAP1/TAZ in HPV-negative cancers. Consistent with this, *PTPN14* is deleted in 1.4% and truncated in an additional 1.6% of HPV-negative HNSCC. *PTPN14* and other many other YAP1/TAZ inhibitors trend towards more frequent mutation in HPV- cancers (Figure 1-Supplement 1). We have updated the text accordingly (lines 121-124).

2. Figure 1: There is no Part E in the figure but is indicated in the Legend. In Figure 1D, the PCNA staining is very difficult to see, even in the insets.

The figure legend has been corrected (lines: 642-644). The contrast for the PCNA panels of Figure 1D was increased to improve visibility of PCNA expression. Adjustments were applied identically to all images. Few cells express PCNA in the HFK and HFK HPV16 E6 cultures.

3. Figure 2 parts C-F: It would be easier to compare the effect on basal vs. suprabasal cells if the basal graphs were next to each other and suprabasal graphs were similarly presented side-by-side.

Figure 2 was updated as suggested.

4. The authors found that E7 activates YAP1 primarily in basal cells. Has this been explored in 2D culture under conditions that induce differentiation? Does YAP1 activation decrease upon differentiation in these 2D experiments like what is seen in rafts? Such data would further strengthen their findings that YAP1 activation is specific to basal cells.

These are excellent suggestions. Indeed, we aim to establish 2D culture conditions that measure the effects of E7 on YAP1. A technical challenge is that growth on plastic potently promotes YAP1 nuclear localization, leaving little room to measure the increased activation of YAP1 downstream of E7. In addition, high cell density (as in our 3D, but not in our test 2D experiments) directs YAP1/TAZ to the cytoplasm in the absence of E7. We assessed many experimental systems but so far concluded that growth in 3D culture in which keratinocytes are grown on devitalized dermis and allowed to undergo stratification best allows us to test the physiological effects of E7 on YAP1.

5. Regarding 'persistence': Have the authors explored whether a mutant HPV18 genome lacking E7 can replicate and/or be maintained over several passages in PTPN14KO cells? Likewise, does a genome carrying the R84S mutation persist in keratinocytes? These experiments would help provide key evidence to support the role of E7-mediated PTPN14 degradation/YAP1 activation in HPV persistence/maintenance.

It is interesting to speculate about whether HPV genome maintenance (thought to depend on E2) and basal cell persistence (our data support a role for E7) are linked or separable events during HPV persistence. Our future studies will aim to address this question.

6. Figure 3A/B: Can the authors provide any additional information about what differentiates the Basal Cell III population from I, II, and IV? In other words, are there other gene expression patterns that define the PTPN14-enriched basal cells?

The Basal-III subset of basal cells were described in Wang et al., 2021 as a transcriptionally distinct non-proliferating subset that is marked by COL17A1 expression. Their data suggest that the basal-III subset is one of the two subsets of basal cells that differentiate directly into suprabasal spinous cells. These additional details were added to the Results section (lines: 181-185).

7. Figure 6: were statistical comparisons performed?

Two-way ANOVAs were performed for each cell line plus/minus doxycycline. Figure 6 and its figure legend were updated accordingly (lines: 709-711). TEADi had a statistically significant inhibitory effect on the growth of HPV16 E7 and HPV18 expressing cells. No statistically significant effect was observed in the GFP expressing primary cells.

8. Line 200-201: is it KRT10 or KRT1 that is being analyzed? Line 200 indicates KRT10, but line 201 indicates KRT1.

KRT10 expression was measured in experiments in Figure 4—figure supplement 1. KRT1 was measured in experiments in Figure 4. The text was altered to clarify this point (lines: 208-210)

9. Citations needed:a. Line 131: statement that PCNA1 increase upon RB1 inactivation

The following citations were added to the text: Cheng et al., 1995; Flores et al., 2000; Lee et al., 1995 (lines: 133).

b. Line 375-376: "previously, HPVE7 was believed to be primarily required to establish a cellular environment conducive to HPV DNA replication in suprabasal cells"

The following citations were added to the text: Cheng et al., 1995; Collins et al., 2005; Flores et al., 2000; McLaughlin-Drubin et al., 2005 (now lines: 403-404).